# Exposure to Proton Pump Inhibitors and the Risk of Incident Asthma in Patients with Coronary Artery Diseases: A Population-Based Cohort Study

**DOI:** 10.3390/jpm12050824

**Published:** 2022-05-19

**Authors:** Tsung-Kun Lin, Chin-Feng Tsai, Jing-Yang Huang, Lung-Fa Pan, Gwo-Ping Jong

**Affiliations:** 1Department of Pharmacy, Taoyuan Armed Forces General Hospital, Taoyuan 32551, Taiwan; ray1007@gmail.com; 2School of Pharmacy, National Defense Medical Center, Taipei 11490, Taiwan; 3Division of Cardiology, Department of Internal Medicine, Chung Shan Medical University Hospital, Taichung 40201, Taiwan; alberttsai54@hotmail.com; 4School of Medicine and Institute of Medicine, Chung Shan Medical University, Taichung 40201, Taiwan; wchinyang@gmail.com; 5Department of Medical Research, Chung Shan Medical University Hospital, Taichung 40201, Taiwan; 6Graduate Institute of Radiological Science, Central Taiwan University of Science and Technology, Taichung 40653, Taiwan; panlungfa@gmail.com; 7Department of Cardiology, Taichung Armed Forces General Hospital, Taichung 41168, Taiwan

**Keywords:** proton pump inhibitors, coronary artery disease, asthma, peptic ulcer disease

## Abstract

We aimed to determine the association between proton pump inhibitor (PPI) use and incident asthma in patients with coronary artery disease (CAD). This nationwide cohort study collected claims data from the Taiwanese Bureau of National Health Insurance from 2004 to 2013. The primary outcome, i.e., the risk of incident asthma, was assessed by estimating hazard ratios (HRs) and 95% confidence intervals (CIs). The adjusted HR of asthma development was estimated using the Cox regression model. Sensitivity and subgroup analyses were also conducted. A total of 8894 PPI users and 12,684 H2-receptor antagonist (H2RA) users were included in patients with CAD. Compared with H2RA use, an increased risk of incident asthma was found between PPI use and the risk of incident asthma in patients with CAD after adjusting for sex, age, urbanization, and low income (HR: 1.41; 95% CI: 1.04–1.89). The sensitivity analysis results were consistent with the main analysis results. However, the subgroup analysis revealed no association of incident asthma in patients with diabetes mellitus, hyperlipidemia, stroke, allergic rhinitis, pneumonia, cancer, or depression in the PPI group compared with those in the H2RA group. In conclusion, PPI use increased the risk of asthma development in patients with CAD.

## 1. Introduction

Asthma is a major noncommunicable disease characterized by recurrent attacks of breathlessness and wheezing, which vary in severity and frequency from person to person. Globally, more than 339 million cases of asthma were reported in 2016 [1]. It is a common disease among children. In 2016, the World Health Organization estimated an annual total of 417,918 deaths due to asthma at the global level and 24.8 million disability-adjusted-life-years attributable to asthma [2]. Patients with asthma and concomitant coronary artery disease (CAD) may be a higher overall healthcare burden than those with either asthma or CAD alone.

Proton pump inhibitors (PPIs) are the primary agents for treating gastroduodenal ulcers and gastroesophageal reflux disease (GERD) and are effective in preventing nonsteroidal anti-inflammatory drugs/aspirin-associated peptic ulcers and ulcer bleeding [3]. A study of PPI therapy in adults with asthma resulted in a small, significant improvement in the morning peak expiratory flow rate [4]. However, the magnitude of this improvement is unlikely to be clinically significant. In the absence of a concomitant esophageal syndrome, empirical therapy is more controversial. Through randomized controlled trials (RCTs), several studies have investigated the efficacy of different PPIs on asthma outcomes [5,6,7,8]. Some studies have indicated that symptoms, lung function, or both, can be improved with the treatment of acid reflux [7,8,9], whereas others have demonstrated no measurable improvement with acid suppression [5,10,11].

To date, the association between PPIs and incident asthma in patients with CAD remains controversial. In this regard, this retrospective cohort study aimed to evaluate the association between PPI use and the risk of incident asthma in patients with CAD based on the data from the National Health Insurance Research Database (NHIRD) of Taiwan.

## 2. Materials and Methods

### 2.1. Study Design and Setting

This retrospective cohort study used insurance claims data provided by the Taiwanese Bureau of National Health Insurance (TBNHI) from January 2004 to December 2013. The data from the NHI of Taiwan were used in this study, which is maintained by the National Research Agency for research purposes. The NHI data includes the data of more than 99% of Taiwan’s population. The database contains information on diagnoses, hospitalizations, examinations, and prescriptions.

Patients with peptic ulcer disease (ICD-9: 041.86, 531–533) and CAD (ICD-9-CM code 410–414) (*n* = 25,082) who were diagnosed between January and December 2004 were identified. The index date was on the date of the first prescription for PPI between January 1, 2004 and December 31, 2012. At least one of the following inclusion criteria had to be met: (1) two or more outpatient visits within 6 months, (2) all prescriptions of PPI were continuously administered to the patients for more than 6 months within a 10-year follow-up period, or (3) one or more outpatients with a diagnosis of peptic ulcer disease and CAD.

### 2.2. Study Outcome

The primary endpoint was the development of asthma, which was defined by the time an asthma (ICD-9: 493) code first appeared in the outpatient claim records. We followed the PPI users or H2-receptor antagonist (H2RA) since their first prescription of PPI or H2RA, until the occurrence of asthma or the end of the study (December 2013).

### 2.3. Study Variables

Comorbidity was defined as any diagnosis within a year after the index date. Asthma-related comorbidities of patients with hypertension were identified using ICD-9-CM codes 401–405, and hyperlipidemia was identified with ICD-9-CM code 272. Other selected conditions included diabetes mellitus (ICD-9-CM code 250), stroke (ICD-9-CM codes 430–438), allergic rhinitis (ICD-9-CM codes 477), pneumonia (ICD-9-CM codes 480–486), cancer (ICD-9-CM codes 140–208), and depression (ICD-9-CM codes 296, 300, 309, and 311). Patients who had a prior history of asthma before the index date were excluded from the study. Finally, the study group comprised 8499 participants with peptic ulcer disease and CAD who were PPI users, and the control group included 12,684 participants with peptic ulcer disease and CAD who were H2RA users (Figure 1).

### 2.4. Data Analysis

Differences in demographic and clinical characteristics between PPI and H2RA users were examined using t-test for continuous variables, whereas chi-square tests were used for categorical variables. The asthma-free survival rates in the two groups were calculated using the Kaplan–Meier method and the log-rank test. The Cox proportional hazard regression model was used to compare the development risk of asthma between PPI and H2RA users. Adjusted HRs and 95% CIs were calculated. Their values were adjusted for important risk factors, such as sex, age, urbanization, and low income, for asthma development.

We also conducted several sensitivity analyses to test the robustness of our primary findings. Initially, propensity score matching (1:1) and inverse probability of treatment weighting (IPTW) were conducted between the two groups using the SAS software. The PSMATCH procedure was used to perform greedy nearest neighbor matching. Logistic regression was performed to estimate the logit probability of exposure to PPI for each individual.

In addition, we conducted subgroup analyses stratified by sex, age, urbanization, presence of hypertension, diabetes mellitus, hyperlipidemia, stroke, allergic rhinitis, pneumonia, cancer, or depression at baseline for the primary outcomes of incident asthma. Statistical significance was considered at *p* < 0.05. All statistical calculations were performed using statistical analysis software, version 9.3 (SAS Institute, Inc., Cary, NC, USA).

## 3. Results

### 3.1. Characteristics of the Study Participants

We identified 25,082 patients with peptic ulcer disease and CAD, of whom we excluded 2949 who had asthma before 2004. Thus, we assessed data for 8499 PPI users and 12,684 H2RA users (controls). The characteristics at baseline are shown in Table 1. Most subjects were 20–60 years of age (69.65% of PPI users and 68.81% of H2RA users). Patients with PPI use had higher rates of comorbid hypertension (27.10% versus 24.99%), diabetes mellitus (13.95% versus 10.67%), stroke (8.38% versus 5.22%), pneumonia (4.47% versus 2.08%), and cancer (6.18% versus 3.50%) than the H2RA users (all *p* < 0.05) at baseline. Inversely, patients with PPI use had lower rates of comorbid hyperlipidemia, allergic rhinitis, and depression. Simultaneously, PPI users used more oral nonsteroid anti-inflammatory drugs, aspirins, statins, beta-blockers, calcium channel blockers, angiotensin-converting enzyme inhibitors, and angiotensin II receptor blockers than those receiving H2RA (Table 1). The follow-up periods were 719,322 person-months in the PPI group and 1,127,352 person-months in the H2RA group.

### 3.2. Association between PPI User and Asthma

Table 2 shows 1887 asthma events (885 for the PPI cohort and 1002 for the H2RA cohort). The incidence was 1.38-fold higher (95% CI, 1.01–1.86) in the PPI cohort than in the H2RA cohort (12.31 versus 8.89 per 10,000 person-months), with an adjusted HR of 1.41 (95% CI, 1.04–1.89) after controlling for sex, age, urbanization, and low income. The Kaplan–Meier curves comparing the cumulative incidence of asthma between the PPI group and the H2RA group were consistent with the main finding (Log-rank test, *p* < 0.001, Figure 2).

### 3.3. Sensitivity and Subgroup Analyses

The trends of results from the sensitivity and subgroup analyses were consistent with the main analysis. The sensitivity analysis for the multiple Cox regression model confirmed the risk of asthma in the patients with PPI use when they were assessed using IPTW and propensity score matching (Table 3). Kaplan–Meier and propensity score matching analysis also showed that the cumulative probability of developing asthma after the 2013 follow-up period showed significant differences between the PPI and H2RA cohorts.

An analysis of the listed comorbidities revealed that asthma prevalence was only higher among patients with hypertension in the PPI cohort (adjusted HR of 1.21; 95% CI, 1.05–1.46) but did not differ among patients without hypertension (adjusted HR of 0.92; 95% CI, 0.78–1.10) (Table 4). Other subgroup analysis revealed no increased risk of incident asthma in patients with or without diabetes mellitus, hyperlipidemia, stroke, allergic rhinitis, pneumonia, cancer, or depression in the PPI group compared with those in the H2RA group.

## 4. Discussion

This large population-based cohort study found a 1.41-fold risk of asthma development after adjusting for age, sex, urbanization, and low income in patients with CAD. The sensitivity analysis results were consistent with the main analysis results. However, compared with H2RA use, PPI use in patients with diabetes mellitus, hyperlipidemia, stroke, allergic rhinitis, pneumonia, cancer, or depression was not associated with increased risk of incident asthma in the subgroup analysis.

A significantly higher prevalence of asthma has been previously reported in patients using PPIs compared with those not using PPIs, particularly in adults [12,13,14]. Similarly, a recent study in children revealed a higher risk of incident asthma with PPI use compared with non-PPI use after high-dimensional propensity score matching (HR, 1.48; 95% CI, 1.41–1.55) [15]. Wang et al. [15] conducted a population-based registry study in Sweden, which included 17,740 children followed up between January 2007 and June 2016. They used data from nationwide Swedish registries along with healthcare and administrative records on both PPIs and registry-recorded hospital diagnoses of asthma. The study reported a 57% increased risk of incident asthma in children administered PPIs. Although the data did not include information on the diagnosis of asthma from primary care clinics, the study findings were significant and correlated with those obtained in the patients with CAD in our study. The biological mechanisms behind the causal relationship between PPI use and incident asthma remain unclear. It has been suggested that PPI medications interfere with the normal digestion of peptides in the stomach, resulting in a T helper 2 cell-dominant response [16,17,18]. This response is thought to be caused by the preservation of epitopes that are normally degraded following exposure to the acidic environment in the stomach. Alternatively, PPIs might directly damage the endothelial function and accelerate the endothelial senescence of the lung [19]. This endocrine disruption hypothesis may explain the current finding of an association of PPI use with an increased risk of asthma.

In the previous clinical observation study, the use of PPI to treat GERD was not associated with incident asthma [7,20,21]. Ruigómez et al. used data from the UK General practice research database and showed that patients treated for GERD with PPI had no significant increased risk of developing asthma (RR: 1.2, 95% CI: 0.9–1.6) [20]. However, Havemann et al. and Jaspersen et al. found that the prevalence of asthma had increased in GERD patients treated with PPI [22,23]. Similarly, the users of PPI were exposed to a significantly increased risk of incident asthma than the nonusers of PPI in our study. To date, the association between PPI use for GERD and incident asthma in patients with GERD remains controversial. Further large randomized studies are warranted to examine the association between PPI use for GERD and incident asthma.

Daily low-dose aspirin is recommended for preventing cardiovascular events in adult patients with CAD, and PPIs are recommended for preventing or treating aspirin-associated gastrointestinal injury [24,25,26]. Previous studies have reported that low-dose aspirin therapy protects against incident asthma [27,28]. However, the present study suggests that PPI treatment increases the risk of asthma development in patients with CAD. Therefore, in clinical practice, the risk of incident asthma should be considered when weighing the benefits and risks of PPI and aspirin treatments in adult patients with CAD.

A major strength of this study is that this was the largest population-based study to examine the association between PPI use and subsequent asthma development in adults. The national database used herein comprises a representative cohort of 1,000,000 people insured by the Taiwan NHI program, and the 10-year observation period ensured the power of our statistical analyses. This is also the first population-based study to suggest that PPI use is associated with an increased risk of asthma development in adult patients with CAD, although some risk factors for PPI are associated with asthma development.

This study has some limitations. First, the data of all the patients in this study were collected from claim datasets of Taiwan NHI that were submitted by primary care clinics. Risk factors for asthma, such as body mass index, smoking status, family history, treatment adherence, environmental tobacco smoke exposure, and diet, were not available from these secondary data. However, considering that we used population-based data, we assumed that no differences existed between PPI users and H2RA users. Second, all the patients in our study had been diagnosed with CAD and peptic ulcer and had received PPI treatment. However, dosing, treatment adherence, and the severity of CAD with acid-related gastrointestinal tract disorders may have differed across patients using different PPIs. Thus, the association between PPI therapy and the risk of asthma may not reflect the effect of the prescribed drugs but rather the severity of the patients’ diseases and treatment adherence.

To conclude, this large population-based cohort study showed PPI use increased the relative risk of incident asthma in patients with CAD. However, there was no association between PPI use and incident asthma in patients with diabetes mellitus, hyperlipidemia, stroke, allergic rhinitis, pneumonia, cancer, or depression. Future studies may help determine whether PPI treatment of at-risk patients with CAD could prevent the development of asthma in clinical practice.

## Figures and Tables

**Figure 1 jpm-12-00824-f001:**
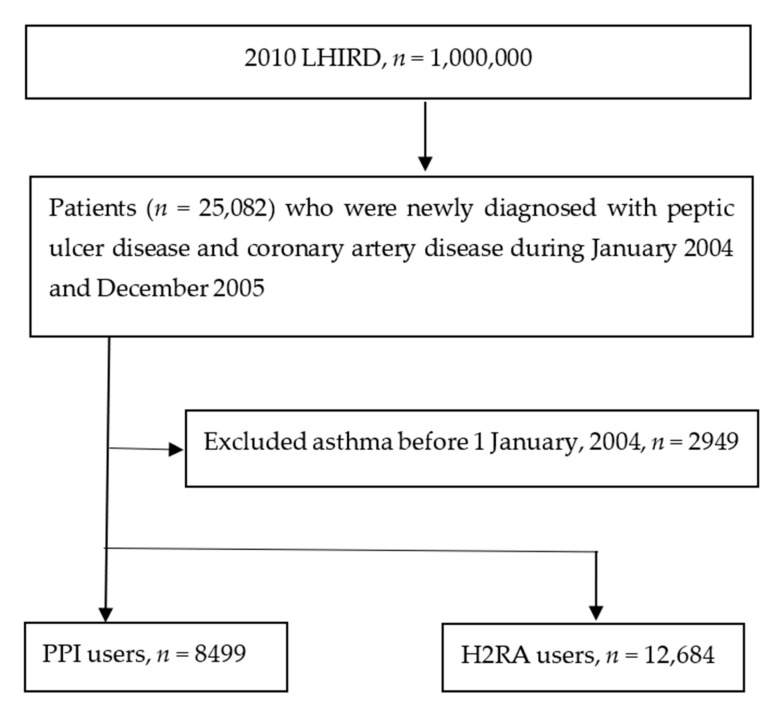
Study flow chart.

**Figure 2 jpm-12-00824-f002:**
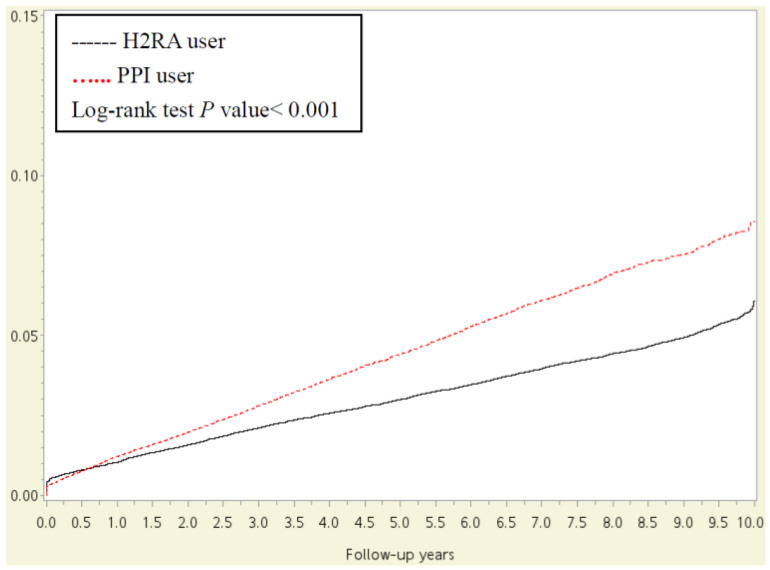
The cumulative probability of new onset asthma in patients with CAD.

**Table 1 jpm-12-00824-t001:** The baseline characteristics among all patients.

	H2RA *n* = 12,684	PPI *n* = 8499	*p*-Value
Sex			<0.001
Female	7013 (55.29%)	3537 (41.62%)	
Male	5671 (44.71%)	4962 (58.38%)	
Age			<0.001
<20	483 (3.81%)	133 (1.56%)	
20–45	4724 (37.24%)	3061 (36.02%)	
45–60	4004 (31.57%)	2858 (33.63%)	
60–75	2466 (19.44%)	1567 (18.44%)	
>= 75	1007 (7.94%)	880 (10.35%)	
Urbanization			0.0736
Urban	7359 (58.02%)	5060 (59.54%)	
Sub-urban	3827 (30.17%)	2593 (30.51%)	
Rural	1498 (11.81%)	846 (9.95%)	
Low income	88 (0.65%)	49 (0.58%)	0.4904
Comorbidity			
Hypertension	3170 (24.99%)	2303 (27.10%)	0.0229
Diabetes mellitus	1353 (10.67%)	1186 (13.95%)	<0.001
Hyperlipidemia	1896 (14.95%)	1254 (14.75%)	0.6901
Stroke	662 (5.22%)	712 (8.38%)	<0.001
Allergic rhinitis	1508 (11.89%)	835 (9.82%)	<0.0001
Pneumonia	264 (2.08%)	380 (4.47%)	<0.001
Cancer	444 (3.50%)	525 (6.18%)	<0.001
Depression	2306 (18.18%)	1462 (17.20%)	0.0912
Concurrent medication			
NSAIDs Aspirins Statins	3567 (28.12%) 5959 (46.98%) 4830 (38.08%)	3723 (43.81%) 4086 (48.08%) 3639 (42.82%)	<0.001 0.0597 <0.001
Beta- blockers	2395 (18.88%)	2506 (29.48%)	<0.001
CCBs	3119 (24.59%)	3127 (36.79%)	<0.001
ACEIs	10,351 (8.16%)	1315 (15.47%)	<0.001
ARBs	1853 (14.61%)	2416 (28.43%)	<0.001

ACEIs: angiotensin-converting enzyme inhibitors; ARBs: angiotensin II receptor blockers; CCBs: calcium channel blockers; H2RA: H2-receptor antagonist; NSAIDs: nonsteroid anti-inflammatory drugs; PPI: proton pump inhibitor.

**Table 2 jpm-12-00824-t002:** Incidence of asthma in study groups.

	H2RA *n* = 12,684	PPI *n* = 8499
Follow up person months	1,127,352	719,322
Event of asthma	1002	885
Incidence rate * (95% C.I.)	8.89 (7.32–10.78)	12.31 (9.95–15.23)
Crude HR (95% C.I.)	Reference	1.38 (1.01–1.86)
aHR (95% C.I.)	Reference	1.41 (1.04–1.89)

* Incidence rate, per 10,000 person-months. aHR: adjusted hazard ratio, the covariates including sex, age, urbanization, low income, and comorbidities.

**Table 3 jpm-12-00824-t003:** The sensitivity analysis for the hazard ratio of study events.

Model	HR (95% CI)	*p*-Value
IPTW ^1^	1.40 (1.03–1.88)	0.0109
Propensity score matching	1.37 (1.04–1.81)	0.0019

^1^ IPTW: inverse probability of treatment weighting.

**Table 4 jpm-12-00824-t004:** Subgroup analysis.

Incidence Rate of Asthma			
	H2RA	PPI	aHR (95% CI)
Hypertension			p for interaction = 0.2992
Without	4.49 (4.07–4.95)	4.13 (3.62–4.70)	0.92 (0.78–1.10)
With	8.49 (7.49–9.62)	10.30 (8.97–11.84)	1.21 (1.05–1.46)
Diabetes mellitus			p for interaction = 0.5962
Without	5.06 (4.64–5.50)	4.94 (4.42–5.52)	0.98 (0.83–1.14)
With	8.84 (7.3–10.71)	8.63 (6.87–10.84)	0.98 (0.67–1.23)
Hyperlipidemia			p for interaction = 0.4678
Without	5.21 (4.79–5.66)	4.95 (4.42–5.54)	0.98 (0.85–1.13)
With	7.25 (6.10–8.62)	7.89 (6.36–9.78)	1.09 (0.82–1.44)
Stroke			p for interaction = 0.5355
Without	5.17 (4.77–5.6)	4.84 (4.34–5.39)	0.94 (0.82–1.15)
With	12.07 (9.45–15.42)	13.43 (10.45–17.26)	1.09 (0.76–1.57)
Allergic rhinitis			p for interaction = 0.9168
Without	5.02 (4.61–5.47)	5.00 (4.48–5.57)	1.00 (0.87–1.15)
With	8.67 (7.28–10.33)	8.79 (6.88–11.24)	1.01 (0.74–1.33)
Pneumonia			p for interaction = 0.7532
Without	5.29 (4.88–5.71)	5.08 (4.57–5.63)	0.96 (0.83–1.11)
With	14.21 (10.01–20.10)	16.12 (11.4–22.8)	0.95 (0.55–1.58)
Cancer			p for interaction = 0.1361
Without	5.41 (5.00–5.85)	5.41 (4.89–5.99)	1.02 (0.89–1.16)
With	7.72 (5.22–11.36)	4.70 (2.83–7.80)	0.60 (0.30–1.21)
Depression			p for interaction = 0.4483
Without	5.11 (4.65–5.54)	4.95 (4.42–5.55)	0.97 (0.81–1.11)
With	7.31 (6.25–8.55)	7.44 (6.07–9.13)	1.08 (0.84–1.41)

## Data Availability

All relevant data are within the article. Further details may be obtained from the corresponding author upon a reasonable request.

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
