# Peer review of "Exposure to Proton Pump Inhibitors and the Risk of Incident Asthma in Patients with Coronary Artery Diseases: A Population-Based Cohort Study"

_jpm, 2022, doi:10.3390/jpm12050824_

Round 1

Reviewer 1 Report

Dear authors, I congratulate you on the paper, it describes a clean and fairly conducted retrospective study. However, I have some concerns to expose to you:

  • It is unclear if the patients taking PPI for the peptic disease had or not associated GERD.
  • In the Discussions, a minimum of one paragraph should be dedicated to the controversies regarding the use of PPIs in managing asthma by managing GERD (as described in the Introduction).
  • I would love to see if the authors can find some new insights on this topic, like newer references (more from 2020-2022) - I feel for this topic 24 references are scarce.

Good luck!

Author Response

Dear authors, I congratulate you on the paper, it describes a clean and fairly conducted retrospective study. However, I have some concerns to expose to you:

It is unclear if the patients taking PPI for the peptic disease had or not associated GERD.

ANS: Thank you for your comment! We include that the prescriptions of PPI for peptic ulcer with or without GERD were continuously administered to the patients for more than 6 months.

In the Discussions, a minimum of one paragraph should be dedicated to the controversies regarding the use of PPIs in managing asthma by managing GERD (as described in the Introduction).

ANS: Thank you for your comment! We had added one paragraph regarding the use of PPIs in managing asthma by managing GERD as below:

In the previous clinical observation study, the users of PPI for treat GERD were not associated with incident asthma [7, 20-21]. Ruigómez et al. using data from the UK General practice research database and showed that patients with GERD treat by PPI had no significant increased risk of developing asthma (RR: 1.2, 95% CI: 0.9-1.6) [20]. However, Havemann et al. and Jaspersen et al. found that the prevalence of asthma had increased in patients with GERD treat by PPI [22, 23]. Similarly, the users of PPI had a significantly increase risk of incident asthma than the nonusers of PPI in our study. To date, the association between PPI use for GERD and incident asthma in patients with GERD remains controversial. Further large randomized studies are warranted to examine the association between PPI use for GERD and incident asthma.

I would love to see if the authors can find some new insights on this topic, like newer references (more from 2020-2022) - I feel for this topic 24 references are scarce.

ANS: Thank you for your comment! We had added the more 4 references in our manuscript.

Reviewer 2 Report

Dear Authors,

I congratulate you for the fine and extensive work regarding the PPI and the risk of asthma in CAD patients. The work is of interest, and you produced a high quality work that could be useful in clinical practice.

I would like to know how you can exclude that asthma is rather a random event that appears in patients with CAD and has no link with PPI, as many patients with CAD are on PPi to prevent digestive haemorrhagic events.

Yours sincerely,

Author Response

I congratulate you for the fine and extensive work regarding the PPI and the risk of asthma in CAD patients. The work is of interest, and you produced a high quality work that could be useful in clinical practice.

I would like to know how you can exclude that asthma is rather a random event that appears in patients with CAD and has no link with PPI, as many patients with CAD are on PPi to prevent digestive haemorrhagic events.

ANS: Thank you for your comment! We only exclude that the prescriptions of PPI were not continuously administered to the patients for more than 6 months.

This manuscript is a resubmission of an earlier submission. The following is a list of the peer review reports and author responses from that submission.

Round 1

Reviewer 1 Report

In the manuscript titled “Exposure to proton pump inhibitors and the risk of incident asthma in patients with coronary artery diseases: A population-based cohort study”, the authors revealed PPI use increased the risk of asthma development in patients with CAD in a population-based cohort study. PPI use increased the risk of asthma development in patients with CAD, this is a risk of adverse events reported firstly in CAD patients using PPI. Although previous articles have reported an association between PPI use and risk of asthma in children. A large sample size, and significance level, have clinical significance, providing reference values for clinicians in PPI use during CAD treatment. The current form already meets the requirements for publication.